# Tensile Strength and Dispersibility of Pulp/Danufil Wet-Laid Hydroentangled Nonwovens

**DOI:** 10.3390/ma12233931

**Published:** 2019-11-27

**Authors:** Chao Deng, R. Hugh Gong, Chen Huang, Xing Zhang, Xiang-Yu Jin

**Affiliations:** 1Engineering Research Center of Technical Textiles, Ministry of Education, College of Textiles, Donghua University, Shanghai 201620, China; hc@dhu.edu.cn (C.H.); tulip_90@163.com (X.Z.); 2Textile Technology, School of Materials, The University of Manchester, Manchester M13 9PL, UK; hugh.gong@manchester.ac.uk

**Keywords:** tensile strength, dispersibility, nonwovens, wet-laid, hydroentanglement.

## Abstract

Wet-laid hydroentangled nonwovens are widely used for disposable products, but these products generally do not have good dispersibility and can block sewage systems after being discarded into toilets. In this study, both pulp fibers and Danufil fibers are selected as we hypothesize that the high wet strength and striated surface of Danufil fibers would allow us to produce nonwovens with better dispersibility while having enough mechanical properties. The wet strength and dispersibility of nonwovens are systematically studied by investigating the influence of the fiber blend ratio, fiber length, and water jet pressure. The results indicate that the percent dispersion could be as high as 81.3% when the wet strength is higher than 4.8 N, which has been improved greatly comparing the percent dispersion of 67.6% reported before.

## 1. Introduction

Due to their softness and high-water absorbency, hydroentangled nonwovens have been used in a wide range of areas, such as personal hygiene products, medical care, moist toilet tissues, and household wiping products [1,2,3,4,5,6,7,8]. However, the traditional disposable products are prone to clogging and blocking household pipes and waste-water systems after being discarded into toilets [9,10]. To avoid these problems, new hydroentangled nonwovens with better dispersibility are needed.

The combination of wet-laid and hydroentanglement techniques offers possibilities to produce dispersible wipes with sufficient wet strength [11,12]. The wet-laid technique is similar to conventional papermaking processes, which uses stapled or chopped synthetic fibers (up to 35 mm), rather than pure pulp fibers. It can produce homogeneous nonwoven textile products [13,14,15,16,17]. The hydroentanglement technique is one of the mechanical bonding techniques that uses fine high-pressure water jets that strike a web to rearrange and entangle fibers, capable of producing nonwovens with good wet strength and softness and high-water absorbency [18,19,20,21]. The wet strength and dispersibility of wet-laid hydroentangled nonwovens depend on raw fiber materials and process parameters, such as fiber length and water jet pressure [22]. For example, Viazmensky et al. [23] adopted relatively low pressure water jets to fabricate the polyester fiber-wood pulp nonwovens. It has been demonstrated that the fiber distribution of obtained sheet material is uniform, and the material strength is improved compared to those fabricated by prior art hydroentanglement processes, which take over 300–2000% of the input energy of entanglement in this process [23]. In addition, Gilmore et al. found that water jets can change the orientation of some of the fibers making up the sheet, and some of them are partly oriented in the thickness direction [24]. Haeubl et al. reported that hydroentangled nonwovens containing Lyocell fibers provide a skin-friendly feeling for adult incontinence [25]. In particular, our group has reported that pulp/Tencel nonwovens fabricated by wet-laid and hydroentanglement techniques could simultaneously have good dispersibility and enough wet mechanical properties [11,26].

Pulp fibers are often used in wet-laid techniques due to their high-water absorbency and degradable properties. In particular, the pit holes in the fiber wall and lumens of pulp fibers could allow water to enter the fiber internal structures. In addition, pulp fibers consist of cellulose and hemicellulose, which are hydrotropic substances and thus are an attribute toward excellent water absorption properties. It should be noted that Danufil fibers have a special cross-section and high wet strength. The crenellated fiber surface not only allows for rapid water penetration during flushing, but can also be attributed to the high surface to volume ratio, and thus enhances the contact area between water and the fiber surface, and has improved water flow resistance during flushing, so better dispersion properties can be expected.

In this study, we hypothesize that the high wet strength and striated surface of Danufil fibers would endow the fabricated nonwovens with better dispersibility while having enough mechanical properties. In brief, pulp/Danufil wet-laid hydroentangled nonwovens are fabricated by using different Danufil fiber lengths, pulp/Danufil fiber blend ratios, and water jet pressures. The average wet strength and dispersibility of nonwovens are analyzed by tensile and disintegration tests. The applications of pulp/Danufil wet-laid hydroentangled nonwovens are compared with those of pulp/Tencel wet-laid hydroentangled nonwovens and bathroom tissue.

## 2. Materials and Methods 

### 2.1. Raw Materials

Pulp fibers (E_pulp_ = 389.6 cN tex^−1^) and Danufil fibers (E_Danufil_ = 389.6 cN tex^−1^) were provided by Canfor Corporation (Prince George, B.C., Canada) and Kelheim Fibers GmbH (Kelheim, Germany), respectively. The fiber length and brightness of pulp fibers (Bleached Kraft Pulp ECF 90) was in the range of 2.4–2.6 mm and 88.5–91.0% (ISO 2470-1: 2016), respectively, while hydrophilic Danufil fibers with different lengths (8, 10, and 12 mm) were used to fabricate wet-laid hydroentangled nonwovens.

### 2.2. Fabrication of Wet-Laid Hydroentangled Nonwovens

The schematic illustration of the fabrication of wet-laid hydroentangled nonwovens is shown in Figure 1a. In brief, pulp fibers and Danufil fibers with different weight ratios (85/15, 75/25, 65/35) were mixed for 20 min (1475 r min^−1^) in the fiber mixing chest to obtain the uniform fiber slurry. The specification of fresh water used here was equivalent to drinking water quality (degree of hardness: 4–6° dH, solid content: < 5 ppm, and the concentrations of chlorides, iron, and manganese: lower than 40, 0.2, and 0.05 mg L^−1^, respectively). The concentration of fiber slurry was finally diluted to 0.4 g L^−1^ before wet-laid formation. The fiber webs were then formed on the inclined mesh belt by dewatering the fiber slurry, which was pumped to the hydroformer (Voith Paper GmbH, Heidenheim, Germany) with a speed of 150 m min^−1^. Next, the fiber webs were transferred and consolidated by hydroentanglement, which involves 5 rows of water jets. Table 1 gives the detailed parameters of the water jet entangling process. Finally, the wet-laid hydroentangled nonwovens were fabricated by dewatering and drying. The drying temperature was 150 ℃, which was supplied by air drum dryers. The details of the nonwoven samples are summarized in Table 2. Furthermore, the nonwoven samples catered to a sustainable development due to the degradable raw materials and environmentally-friendly preparation method (Figure 1b).

### 2.3. Characterization

#### 2.3.1. Morphological Characterization by SEM

The scanning electron microscope (SEM, TM3000, Hitachi, Tokyo, Japan) was adopted to characterize the morphologies of fibers and wet-laid hydroentangled nonwovens. Before SEM observation, the sputter coater (LDM150D, Taiwan Jingda, China) used the gold–platinum alloy to coat the nonwoven samples.

Parameters of raw fiber materials were measured from SEM images by using Adobe Acrobat. The width and thickness of pulp fibers and the diameter of Danufil fibers were calculated and expressed as the average ± standard deviation. At least 100 measurements were taken for each sample.

#### 2.3.2. Characterization of Tensile Strength

The wet tensile strength in the machine direction (MD) and cross direction (CD) was tested by an electronic fabric strength tester (YG026MB, Wenzhou Fangyuan Instrument Co., Ltd., Wenzhou, China), according to the ISO 9073-3: 1989 textiles-test methods for nonwovens [27]. Samples with a moisture content of 200% and size of 50 × 250 mm^2^ were conducted. The experiment parameters were 200 mm (clamp distance) and 100 mm min^−1^ (stretching rate). The average wet strength (AWS) was adopted to represent the overall mechanical properties of nonwoven samples. The AWS is calculated as follows:AWS = (T_MD_ + T_CD_)/2,(1)
where T_MD_ and T_CD_ are the tensile strength in MD and CD directions, respectively.

#### 2.3.3. Measurement of Dispersibility and Wiping Hands Application

For the dispersibility measurement of nonwovens, the mass of samples (50 × 50 mm^2^) was marked as M, and then samples were put into a beaker with a liquid (physiological saline) volume of 600 mL and stirred for 10 min (400 rpm) by a magnetic stirrer (RS-1DN, AS ONE Corporation, Osaka, Japan), according to the guidelines [11,28]. After stirring, a 12.5 mm perforated plate sieve was used to screen the materials in the beaker. Lastly, the residual fragments screened by the sieve were weighted and marked as M_1_. Therefore, the percent dispersion R_d_ is calculated as follows:R_d_ = (M − M_1_)/M × 100%,(2)

The temperature and relative humidity of all the experiments were 20 ± 2 °C and 65% ± 4%, respectively. For every experiment, five samples were measured, and the test results were expressed as the average ± standard deviation. The one-way analysis of variance method, which sets the *p* value at 0.05, was adopted to analyze the experimental data, and data differences were considered statistically significant and marked by * if *F > F*-crit. 

In the application of wiping wet hands, a small kettle with a quantitative extrusion unit containing tap water was used to spray the hands at a distance of 30 cm, and there were 5 extrusions for one experiment to ensure the same water content. After the wet hands were wiped without water, different materials were observed to compare the wiping effects.

## 3. Results and Discussion 

### 3.1. Parameters of Fibres

The SEM images indicate that the width and thickness of the ribbon-like pulp fibers were 34.9 ± 9.5 μm and 5.1 ± 1.9 μm, respectively (Figure 2a,b,d,e), and the diameter of serrated cross-sectional Danufil fibers was 12.0 ± 1.1 μm (Figure 2c,f).

The flexural rigidity of fiber had a high impact on fiber entanglement during nonwoven fabrication. Mao et al. reported that increasing the flexural rigidity of fiber leads to the decrease of fiber entanglement during the hydroentanglement process [29]. To calculate and compare the flexural rigidity of pulp fiber and Danufil fiber, the relative flexural rigidity is defined as follows [30]:R_fr_ = (1/4π) × η_f_ × (E/γ) ×10^−5^,(3)
where R_fr_ is the fiber relative flexural rigidity of fiber material with the same linear density (cN cm^2^ tex^−2^), η_f_ represents the cross-section shape coefficient, which is the actual sectional moment of inertia divided by the equivalent circular sectional moment of inertia with the same area, E is the fiber elastic modulus (cN tex^−1^), and γ is the fiber density (g cm^−3^). In order to calculate the η_pulp_, the cross-section shape of pulp fiber was assumed to be rectangular (Figure 3a), according to the measurement and observation of the SEM images. Therefore, the η_pulp_ can be calculated as follows:η_pulp_ = I_f_/I_0_ = (BH^3^/12)/(π(d_p_)^4^/64),(4)
where I_f_ is the actual sectional moment of inertia of pulp fiber (cm^4^), I_0_ is the equivalent circular sectional moment of inertia with the same area of pulp fiber cross-section (cm^4^), B is the width, H is the thickness of pulp fiber, and d_p_ is the equivalent circular diameter. The values of B and H of pulp fiber were substituted into Equation (4), thus the η_pulp_ was 0.15. In addition, the cross-section shape of Danufil fiber was assumed to be circular (Figure 3b), and hence the η_Danufil_ is 0.75 [30] when γ_pulp_ is 0.70 g cm^−3^ and γ_Danufil_ is 1.37 g cm^−3^. The elastic modulus of pulp fiber (E_pulp_) and Danufil fiber (E_Danufil_) were 389.6 cN tex^−1^ and 644.5 cN tex^−1^, respectively. These parameters were then substituted into Equation (3), and after calculation, the R_pulp_ and R_Danufil_ were 6.44 × 10^−5^ cN cm^2^ tex^−2^ and 2.81 × 10^−4^ cN cm^2^ tex^−2^, respectively. The results illustrate that pulp fiber is more easily bended or entangled due to the smaller R_pulp_ when compared with the R_Danufil_ of Danufil fiber in the same hydroentanglement conditions.

### 3.2. Structure of Wet-Laid Hydroentangled Nonwovens

The micro structures of wet-laid nonwovens before and after hydroentanglement are shown in Figure 1c–e, respectively. In addition, as can be seen in Figure 2g, most Danufil fibers were aligned along the MD direction, which served as the “skeleton” structure of nonwovens, while pulp fibers entangled with themselves and Danufil fibers. This phenomenon could be ascribed to the fact that the ribbon-like pulp fibers were more prone to bend and crimp when struck by water jets due to their smaller relative flexural rigidity. During the process of hydroentanglement, wet-laid fiber web was impinged by fine high-pressure water jets, which dragged pulp and Danufil fibers to travel from the top surface to the bottom and back to the top surface again with the assistance of reflective water jets (Figure 2h,i) [11]. 

### 3.3. Wet Strength of Wet-Laid Hydroentangled Nonwovens

Wet tensile strength is one of the key parameters of wet-laid hydroentangled nonwovens. Figure 4a–c show the relationship between average wet strength and total water jet pressures with different Danufil fiber lengths and contents. It can be seen that the average wet strengths of nonwoven samples N1–N9 increased as the water jet pressures increased from 80 to 250 bars. This can be explained by the fact that fiber entanglements and wrapping angles increased with the increase of water jet pressures.

With regards to the impact of Danufil fiber content on the average wet strength of nonwoven samples, it is clearly shown that increasing the Danufil fiber content with the same fiber length results in the increase of the average wet strength of wet-laid hydroentangled nonwovens (Figure 4d–f). This is attributed to the higher wet tensile strength of Danufil fibers compared to that of pulp fibers.

Comparing the average wet strength of nonwoven samples with different Danufil fiber lengths (Figure 4g–i), the average wet strength under the same pulp/Danufil fiber blend ratio and water jet pressures is sorted as follows: N1 < N2 < N3, N4 < N5 < N6 and N7 < N8 < N9 (Table 3). It is clear that the average wet strength of wet-laid hydroentangled nonwovens increased with the increase of Danufil fiber length. This is related to the fact that longer fiber lengths can lead to more efficient fiber entanglements, which improves the tensile strength of nonwovens after hydroentanglement. These results demonstrate that the increase in Danufil fiber length, content, and water jet pressures improve the average wet strength of wet-laid hydroentangled nonwovens.

According to previous study [31], the average wet strength of moist wiping materials which could be dispersible is required to be higher than 4.8 N. Therefore, nonwoven samples, N8 and N9, can have sufficient wet strength for use and others (N1–N7) can meet the requirement only if the water jet pressure is equal or higher than 130 bars (except sample N1 and N2 at a water jet pressure of 130 bars).

### 3.4. Dispersibility of Wet-Laid Hydroentangled Nonwovens

Dispersibility is an important characteristic which enables wet-laid hydroentangled nonwovens to become potentially flushable [11]. Figure 5 shows the structural schematic of wet-laid hydroentangled nonwovens and the dispersion principle of materials. As shown in Figure 5a, Danufil fibers serve as the “skeleton” structure of nonwovens, entangling with pulp fibers to form different entangled structures (also confirmed by Figure 2g). It can be found that these entangled fibers can form “Z”-like structures, strap-on structures, “X”-like structures, and closed-loop structures (Figure 5b). In addition to endowing materials with mechanical properties, these relatively weak fiber entanglement structures also provide the possibility for dispersion of materials. When the fiber entanglement structures were hit by water flow generated by mechanical agitation in water, the hydrogen bonds between fibers were destroyed by the water molecule, leading to the decrease of bonding intensity between fibers, and then the weak fiber entangling structures were easily disentangled under the impact of the water shearing force (Figure 5c), and hence these materials could become dispersible. Moreover, the striated surface of Danufil fibers allowed water to penetrate during flushing (Figure 5c), thus the materials could disperse more easily. Furthermore, the increased surface-to-volume ratio of Danufil fiber resulted in a higher water flow resistance, leading to more effective dispersion of the nonwoven samples.

Figure 6 indicates the percent dispersion of pulp/Danufil wet-laid hydroentangled nonwoven samples. As the water jet pressure increased, the percent dispersion of all samples decreased, especially when the total water jet pressure was higher than 190 bars (Figure 6a–c). As aforementioned, nonwoven samples dispersed when the water flow penetrated in and disentangled the fiber entanglement structures. As the higher water jet pressure increased the fiber entanglements, the structures of nonwoven samples became compact and stable, and also less water could enter into the materials. Hence, higher water jet pressure naturally resulted in a lower percent dispersion of nonwoven samples. 

Comparing the percent dispersion of nonwoven samples with the same fiber length with water jet pressures of 80 to 130 bars (Figure 6d–f), it can be found that the percent dispersion of materials increased with the increase of Danufil fiber content. This can be ascribed to the fact that the ribbon-like pulp fibers were more likely to bend and crimp, owing to the lower fiber relative flexural rigidity compared to that of Danufil fibers. Thus, it is understandable that under low-water jet pressure, nonwoven samples achieved less fiber entanglements with higher Danufil fiber content. However, the percent dispersion of nonwoven samples presented the opposite trend with the increase of Danufil fiber content when the total water jet pressures were in the range of 190–250 bars (Figure 6d–f). Increasing the Danufil fiber content resulted in tighter fiber entanglements, because both pulp fibers and Danufil fibers could achieve sufficient fiber entanglements to form the fiber entanglement plateaus under high water jet pressure. Therefore, the percent dispersion of nonwoven samples decreased with the increase of Danufil fiber content under the higher water jet pressure (190–250 bars).

The percent dispersion of nonwoven samples for the same fiber blend ratio decreased as the Danufil fiber length increased in the water jet pressure range of 80–190 bars (Table 4). This phenomenon could be explained by considering that the length increase of fiber with the same diameter leads to a decrease of fiber rigidity, thus improving the fiber entanglements of nonwoven samples under the same water jet pressure. However, when the water jet pressure was higher than 190 bars, the percent dispersion decreased sharply and became near-identical (Figure 6g–i). This shows that the Danufil fiber length had little effect on the percent dispersion of wet-laid hydroentangled nonwovens under high water jet pressure. Higher water jet pressure was conductive to increasing fiber entanglements, which may lead to lower dispersibility. Considering the required wet strength of dispersible moist wiping materials, the optimum wet-laid hydroentangled nonwoven was sample N8 (65/35), which had sufficient wet strength (5.2 ± 0.7 N) while having higher dispersibility (81.3% ± 4.1%) with a water jet pressure of 80 bars. 

### 3.5. Applications of Wet-Laid Hydroentangled Nonwovens

Comparisons of the average wet strength and percent dispersion of pulp/Danufil wet-laid hydroentangled nonwovens (sample N8) with pulp/Tencel wet-laid hydroentangled nonwovens and conventional bathroom tissue are shown in Figure 7. As can be seen in Figure 7a, the average wet strength of N8 (0.31 ± 0.02 MPa) was smaller than that of pulp/Tencel wet-laid hydroentangled nonwovens (0.35 ± 0.03 MPa), but the bathroom tissue (0.27 ± 0.02 MPa) was the smallest. In addition, N8 had better dispersibility (81.3% ± 4.1%) than that of pulp/Tencel wet-laid hydroentangled nonwovens (67.6% ± 4.6%) and was comparable to that (82.6% ± 5.1%) of bathroom tissue (Figure 7b). Moreover, the visualized dispersion results of these three materials are shown in Figure 7c–e, respectively. N8 had the best dispersion effects among these three materials. These three materials were used to wipe wet hands to compare the actual use effect, and the wiping results were consistent with the tensile strength experimental data (Figure 7f–h). These results prove that pulp/Danufil wet-laid hydroentangled nonwovens can possess the same application effects with sufficient wet strength while having better dispersibility (Appendix A).

## 4. Conclusions

In the present paper, experiments are performed to investigate the influence of water jet pressure, Danufil fiber content, and fiber length on the average wet strength and dispersibility of wet-laid hydroentangled nonwovens. The increase in water jet pressure, Danufil fiber content, and fiber length leads to the increase in the wet strength of wet-laid hydroentangled nonwovens. The dispersibility of wet-laid hydroentangled nonwovens decreased with increasing water jet pressure and Danufil fiber length, especially when the water jet pressure was higher than 190 bars. Importantly, the percent dispersion increased with the increase in Danufil fiber content under low water jet pressure (80–130 bars), while it presented the opposite trend when the water jet pressure was higher (190–250 bars). The applications of pulp/Danufil wet-laid hydroentangled nonwovens prove that wet-laid hydroentangled nonwovens can have higher dispersibility with enough wet strength by adopting pulp and Danufil fibers. In summary, pulp/Danufil (65/35) wet-laid hydroentangled nonwovens with a Danufil fiber length of 10 mm, fabricated with a total water jet pressures of 80 bars, have been found to offer sufficient wet strength (5.2 ± 0.7 N) with better dispersibility (81.3% ± 4.1%).

## Figures and Tables

**Figure 1 materials-12-03931-f001:**
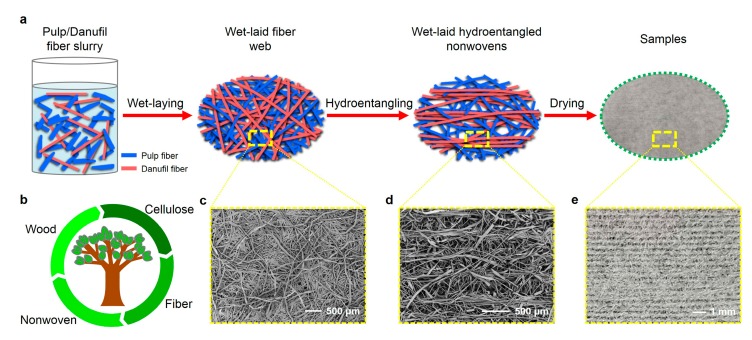
(**a**) Schematic illustration of the fabrication of wet-laid hydroentangled nonwovens; (**b**) biodegradability and environmental prospects of materials; SEM photographs show the micro structures of (**c**) wet-laid fiber webs and (**d**) wet-laid hydroentangled nonwovens; (**e**) optical photograph shows the structure of samples.

**Figure 2 materials-12-03931-f002:**
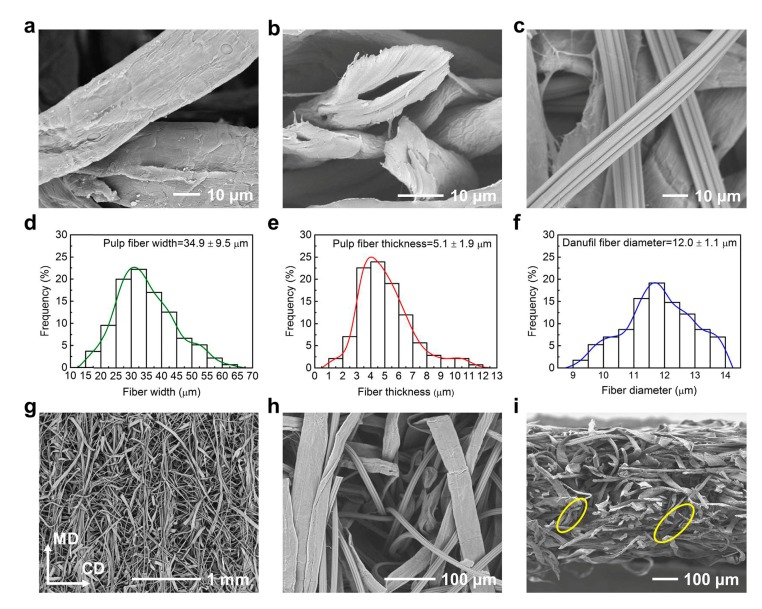
SEM images and parameters of fibers and wet-laid hydroentangled nonwovens (N8). (**a**,**b**) Pulp fibers; (**c**) Danufil fibers; pulp fiber (**d**) width and (**e**) thickness frequency distribution; (**f**) Danufil fiber diameter frequency distribution; (**g**) surface structure; (**h**) water jet pinhole; (**i**) cross-section.

**Figure 3 materials-12-03931-f003:**
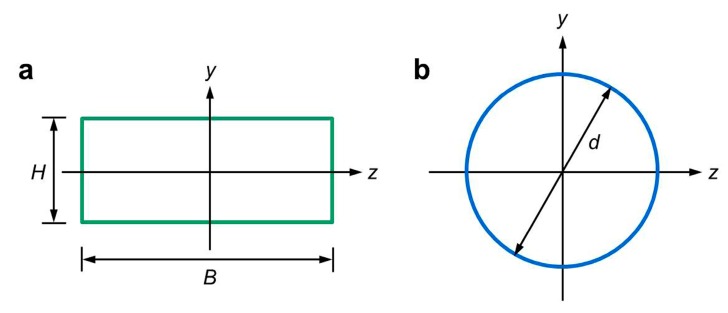
Model diagram for rectangular and circular sectional moment of inertia calculation. (**a**) Pulp fiber; (**b**) Danufil fiber.

**Figure 4 materials-12-03931-f004:**
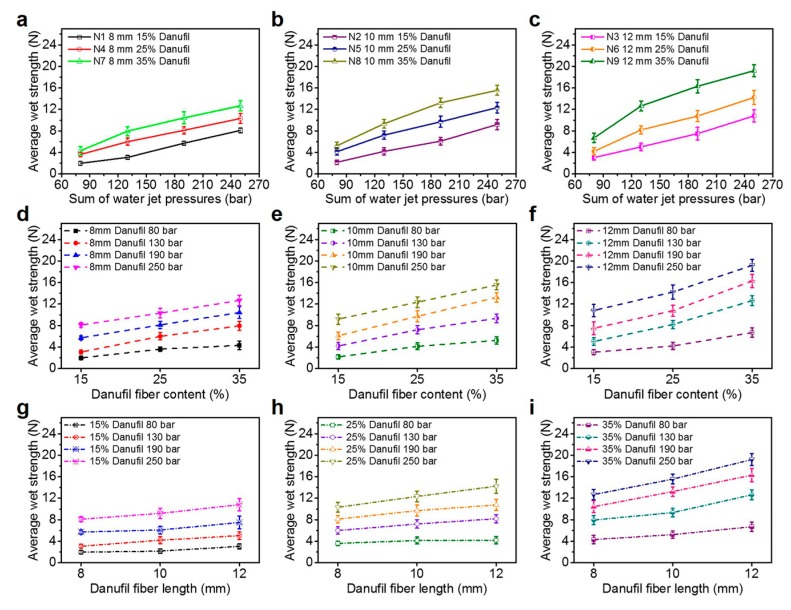
Average wet-strength. (**a**–**c**) Effect of water jet pressure; (**d**–**f**) effect of Danufil fiber content; (**g**–**i**) effect of Danufil fiber length.

**Figure 5 materials-12-03931-f005:**
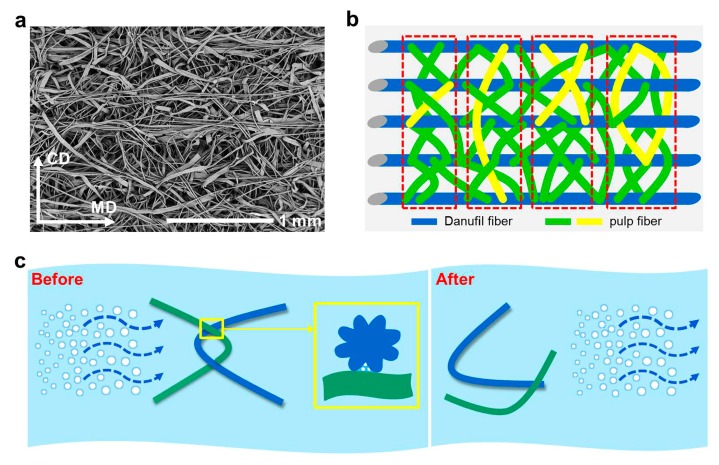
(**a**) Surface SEM image and (**b**) structural schematic of wet-laid hydroentangled nonwovens (N8); (**c**) schematic representation of water flow disentangling the fiber entanglement.

**Figure 6 materials-12-03931-f006:**
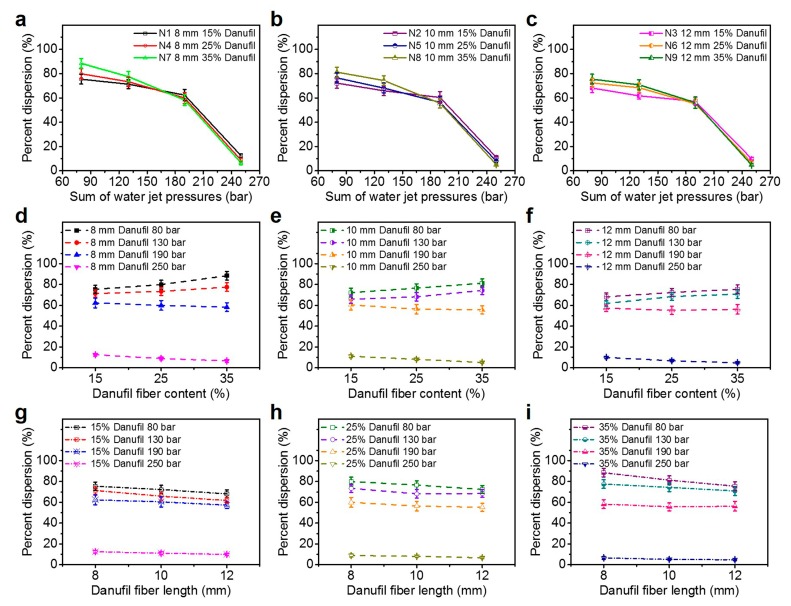
Percent dispersion. (**a**–**c**) Effect of water jet pressure; (**d**–**f**) effect of Danufil fiber content; (**g**–**i**) effect of Danufil fiber length.

**Figure 7 materials-12-03931-f007:**
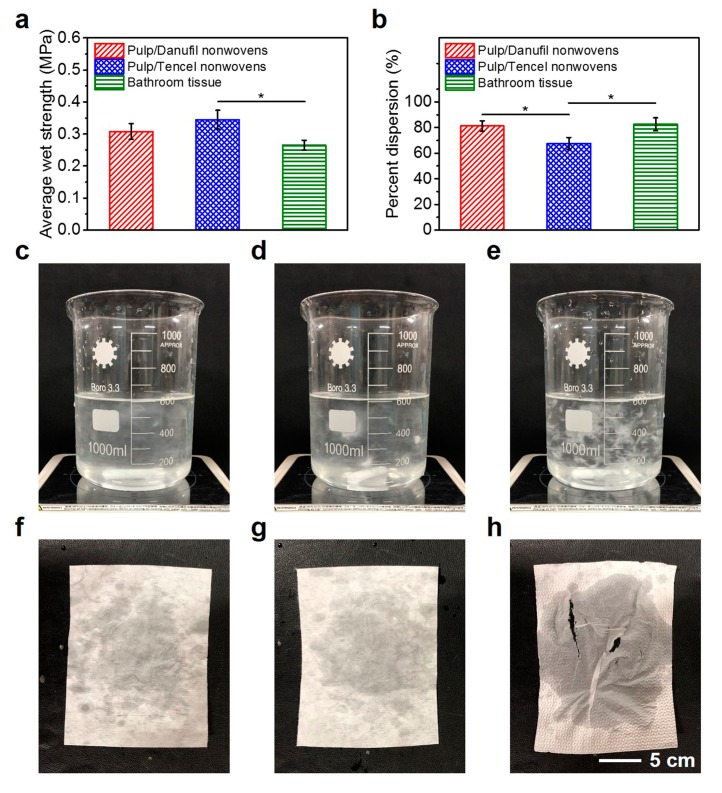
Comparisons of pulp/Danufil wet-laid hydroentangled nonwovens, pulp/Tencel wet-laid hydroentangled nonwovens, and bathroom tissue. (**a**) Average wet strength; (**b**) percent dispersion; (**c**–**e**) dispersion effect; (**f**–**h**) samples after wiping wet hands.

**Table 1 materials-12-03931-t001:** Specific parameters of the hydroentanglement process.

Total Water Jet Pressure (bars)	Specific Pressure Values of Each Row (bars)
80	30, 50
130	30, 50, 50
190	30, 50, 50, 60
250	30, 50, 50, 60, 60

**Table 2 materials-12-03931-t002:** Composition of wet-laid hydroentangled nonwovens.

Samples	Danufil Fiber Length (mm)	Pulp/Danufil Fiber Blend Ratio (%)	Basis Weight (g m^−2^)
N1	8	85/15	65
N2	10	85/15	65
N3	12	85/15	65
N4	8	75/25	65
N5	10	75/25	65
N6	12	75/25	65
N7	8	65/35	65
N8	10	65/35	65
N9	12	65/35	65

**Table 3 materials-12-03931-t003:** Average wet strength of the wet-laid hydroentangled nonwovens tested.

Samples	Average Wet Strength (N)			F-test (P = 0.05)
	80 (bars) (SD)	130 (bars) (SD)	190 (bars) (SD)	250 (bars) (SD)
N1	2.0 ± 0.3	3.1 ± 0.4	5.7 ± 0.4	8.1 ± 0.5
N2	2.2 ± 0.4	4.2 ± 0.6	6.1 ± 0.7	9.2 ± 1.0
N3	3.1 ± 0.5	5.0 ± 0.7	7.5 ± 1.2	10.8 ± 1.1
N4	3.6 ± 0.4	6.0 ± 0.7	8.1 ± 0.7	10.3 ± 0.9
N5	4.1 ± 0.6	7.2 ± 0.8	9.7 ± 1.0	12.3 ± 1.0
N6	4.2 ± 0.7	8.2 ± 0.8	10.8 ± 1.0	14.2 ± 1.3
N7	4.3 ± 0.8	8.0 ± 0.8	10.4 ± 1.1	12.7 ± 1.0
N8	5.2 ± 0.7	9.3 ± 0.8	13.2 ± 0.9	15.6 ± 0.9
N9	6.7 ± 0.9	12.6 ± 0.9	16.3 ± 1.2	19.2 ± 1.1

**Table 4 materials-12-03931-t004:** Percent dispersion of the wet-laid hydroentangled nonwovens tested.

Samples	Percent Dispersion (%)			F-test (P = 0.05)
	80 (bars) (SD)	130 (bars) (SD)	190 (bars) (SD)	250 (bars) (SD)
N1	75.4 ± 4.0	71.3 ± 3.6	62.3 ± 4.7	12.5 ± 1.4
N2	72.2 ± 4.2	65.9 ± 3.9	60.4 ± 4.9	11.1 ± 1.3
N3	68.1 ± 3.8	61.7 ± 2.7	57.2 ± 3.3	9.9 ± 1.1
N4	79.9 ± 4.1	73.4 ± 3.8	60.0 ± 4.6	8.9 ± 1.5
N5	76.5 ± 4.2	68.1 ± 4.0	56.4 ± 4.6	8.1 ± 1.1
N6	72.4 ± 3.8	68.4 ± 3.7	55.2 ± 3.9	6.7 ± 1.0
N7	88.5 ± 4.0	77.5 ± 4.2	58.3 ± 4.3	6.6 ± 1.5
N8	81.3 ± 4.1	74.4 ± 3.9	55.7 ± 3.8	5.1 ± 0.7
N9	75.5 ± 4.1	70.8 ± 4.1	56.2 ± 4.6	4.8 ± 0.9

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
