# Peer review of "Tensile Strength and Dispersibility of Pulp/Danufil Wet-Laid Hydroentangled Nonwovens"

_materials, 2019, doi:10.3390/ma12233931_

Round 1

Reviewer 1 Report

The paper presents results of investigations on hydroentangled nonwovens manufactured from a mixture of pulp and Danufil fibers. The results are interesting and worthy to publish in the journal. Nevertheless, Before publication the paper need major revision.

Comments:

1/ Title

The title of the paper: Enhancing the dispersibility of wet-laid hydroentangled nonwovens by using pulp and Danufil fibers is not accurate and does not describe well the content of the paper.

2/ Introduction

The essential part of Introduction is a combination of abstract of the paper and conclusions. The Introduction does not present clearly the goal of the research

3/ Materials and methods

More information on the pulp fibers should be given.

What is exactly presented in Table 1? Parameters of fibres (p.2. l. 64) or parameters of nonwovens (p.2 l. 76) ?

The kind of pressure (total, specific) and labelling of jet heads in Table 2 is not clear.

Combination of pictures in Figure 1 is confusing. What information gives Fig.1b?

Details of drying should be given.

Adobe Acrobat was really used for determining the geometrical parameters of fibres? What is width and thickness of pulp fiber ? How you determine diameter of non-circular Danufil fibres ?

Which liquid was used in dispersibility tests?

4/ Results and discussion

Presentation of results and essential comments of obtained values are mixed with introductory or general sentences, which add nothing to the discussion.

For example the discussion starts with following sentences:  In our work, wet-laid hydroentangled nonwovens are fabricated by combining the wet-laid formation and hydroentanglement. Pulp fibers are often used in wet-laid techniques due to their high water absorbency and degradable properties (p.4 l.120 – 123). It is known from the beginning of your paper.

The same: In this study, different weight ratios (85/15, 75/25, 65/35) of pulp fibers and Danufil fibers are adopted to fabricate the wet-laid hydroentangled nonwovens (p.5 l. 162-164). You repeat the information given earlier.

What is marked with a yellow circle in the Figure 2i ?

Generally English is very poor and need to be considerably improved.

Reviewer 2 Report

The subject is definitely relevant, and the manuscript is well-written. However, please address the below questions before consideration for publication.

Could you please explain what is novel about this manuscript? What is the knowledge contribution to the scientific community?

The dispersibility of the fiber slurry greatly depends on the surface characteristics of the fibers (oil, finish, hydrophobicity/hydrophilicity, etc.), and the type of the water used (hardness, purity, minerals, etc.). It is difficult to conclude without knowing, and controlling these parameters.

Please cite the reference and standard for the dispersibility test method.

Reviewer 3 Report

The authors have reported an interesting work on improving the current wiping materials by means of the preparation of pulp/Danufil fibre-based hydroentangled nonwoven. I would recommend the acceptance of this research work upon completion of some minor concerns:

Comment 1: The first corresponding author’s e-mail should be replaced with an institutional one.

Comment 2: The missing cross reference in line 38 must be included.

Comment 3: The statement “Our hypothesis that the high wet strength and striated surface of Danufil fibers would allow us to produce nonwovens with better dispersibility while having enough mechanical properties” in lines 49-51 is poorly explained and written. It must be rewritten.

Comment 4: The authors are encouraged to transfer all the results and discussion, namely lines 56-58 and 78-79) to the result section.

Comment 5: Sections 2.3.1 (lines 88-92) and 2.3.2 (lines 93-96) should comprise the same section, as they refer to the same characterization technique (SEM).

Comment 6: The authors should include the whole first part of the results in section 3.1 (lines 120-129) in the introduction, as it does not allude to results, but to the justification of the utilisation of the current raw materials.

Comment 7: How did the authors calculate the elastic moduli of the fibres (lines 153-154)? It has not been included neither in the experimental nor in the result sections.

Comment 8: Scales of the graphs (at least those sharing the raw) in lines 178-180 should be the same to facilitate a proper comparison.

Comment 9: Just as a curiosity. According to what the authors reported, which wet strength would they consider as the maximum above which the dispersibility could experience a dramatical reduction as a consequence of the excessively high entangling forces?

Comment 10: As the authors have reported regarding the dispersion percentages when applying water pressures lower than 130 bars (lines 231-238), the values present confidence intervals that overlap to each other. It should be required to conduct a further post hoc test (such as Scheffé test or similar) in order to determine whether the values can be statistically distinguishable to each other, otherwise the authors should not assure that (for example) at low water jet pressures the dispersion percentages are different, while they remain equal at higher entangling pressures. Moreover, the statement indicating that that percentage decreases with the Danufil fibre content due to its higher relative rigidity opposes to the results obtain when analysing the wet strengths, which should also reduce owing to the same fact.

Comment 11: Did the authors measure any parameter (such as dispersing time, size of the residual pieces of paper, …) to assure that “N8 has the best dispersion effect among these three materials” (line 266)? Was it just a matter of visual inspection?

Comment 12: How do the authors control the “hand wetting process” (lines 266-270)? Did they apply a specific water flow during a controlled wetting time? Otherwise I would say that it is hard to compare the way in which these materials dry somebody’s hands.

Comment 13: The authors should check some minor grammatical errors, reduce some figure captions and avoid some repetitions (such as in lines 162-163) throughout the whole manuscript.

Round 2

Reviewer 1 Report

The correction of the text was made in accordance with my comments.

I suggest shortening the title:

Tensile strength and dispersibility of pulp/Danufil wet-laid hydroentangled nonwovens.

It is sufficient.

Reviewer 2 Report

Thank you for the modifications!